

# Extraction of polysaccharides from *Amaranthus hybridus* L. by hot water and analysis of their antioxidant activity

Yujia Tang[1],[*], Yirong Xiao[2],[*], Zizhong Tang[1], Weiqiong Jin[1], Yinsheng Wang[1], Hui Chen[1], Huipeng Yao[1], Zhi Shan[1], Tongliang Bu[1] and Xiaoli Wang[1]

[1] College of Life Sciences, Sichuan Agricultural University, Ya'an, China
[2] Sichuan Agricultural University Hospital, Ya'an, China
[*] These authors contributed equally to this work.

Corresponding author
Zizhong Tang, 14126@sicau.edu.cn

## ABSTRACT

**Background:** *Amaranthus hybridus* L. is an annual herb that belongs to the Amaranthceae family, a type of multi-purpose grain, vegetable and feed crop that has received considerable attention due to its great economic value. However, the composition of polysaccharides from *A. hybridus* has rarely been previously reported.
**Methods:** In this study, the aboveground part of *A. hybridus* was used as material and polysaccharides were isolated by the hot water extraction method. Two acidic polysaccharides were isolated and purified by the Sevage method and diethylaminoethyl cellulose-32 column chromatography.
**Results:** Two acidic polysaccharides were obtained from *A. hybridus*: AHP-H-1 and AHP-H-2. There were significant differences between the monosaccharide content from each sample according to gas chromatography-mass spectrometer. AHP-H-2 had higher antioxidant activity in vitro than AHP-H-1. The 2,2-diphenyl-1-picrylhydrazyl radical scavenging rate of two mg/mL AHP-H-2 was 80%, its hydroxyl radical scavenging rate was approximately 48.5%, its superoxide anion radical scavenging rate was 85.3% and its reduction ability of $Fe^{3+}$ was approximately 0.92. The total antioxidant capacity of each milligram of AHP-H-2 was 6.5, which was higher than ascorbic acid.
**Conclusion:** The results of the study promote the effective use of *A. hybridus* and provide a theoretical basis for its development.

## INTRODUCTION

*Amaranthus hybridus* L., an annual herb that belongs to the Amaranthus family *Amaranthceae,* is a type of common grain amaranth crop (*Sun, 1998*; *Zhu, 1990*). According to "The Shen Nong Ben Cao Jing" (*Gu, 2004*) records, *A. hybridus* has many effects, such as improving eyesight, relaxing the bowels, dispelling cold, removing heat, etc. Amaranth, containing fiber, protein, tocols, squalene, and the substances possessing cholesterol-lowering function, is a particularly important crop for developing countries

(*Johns & Eyzaguirre, 2007*). Most recently, a comprehensive review was published, which is focusing mainly on health effects, such as hypocholesterolemic activity, influence on the immune system, antitumor effect, action on blood glucose levels, effects on liver functions, hypertension, antienemic effect, antioxidant activity, celiac disease, and antiallergic action (*Caselato-Sousa & Amaya-Farfán, 2012*). The carbohydrate content among the various components of the grain is the highest and can reach up to 60%.

Previous studies have suggested that plant polysaccharides have a wide range of biological activities, such as antioxidant (*Yang et al., 2014*; *Tahmouzi & Ghodsi, 2014*; *Han et al., 2015*) anti-tumour, immunoregulation (*Kim, Kang & Kim, 1990*), hypoglycemic and hyperlipidemic (*Jin et al., 2013*; *Chen et al., 2013*; *Tang et al., 2013*; *Xu et al., 2016*; *Liu et al., 2002*; *Liao et al., 2016*) and so on. Polysaccharides usually form conjugates with proteins, polyphenols and lipids. It has been reported that the protein components in polysaccharides have an effect on the free radical scavenging activity of some parts of the polysaccharides. The free radical scavenging activities of exopolysaccharides studied in the literature were significantly correlated with protein content. Phenolic compounds, especially phenolic acids, play an important role in free radical scavenging of xylan and xylo-oligosaccharides. Therefore, the degree of complexation of polysaccharides with total phenols or proteins may indicate their strong antioxidant capacity. The phenolic and protein contents of pectin and other complex carbohydrates on the market were analyzed. It was found that the phenolic content of lipopolysaccharides, pectin and polygalacturonic acid was higher, the protein content of xylan and pectin was higher, and their carbohydrate content was basically higher. The experimental results showed that pectin and polygalacturonic acid had stronger antioxidant ability and antioxidant ability. The chemical ability may be attributed to phenols and proteins (*Hu, 2016*). Research into the grain amaranth is now mainly focused its exploitation as a forage crop and use in healthcare, but polysaccharides from amaranth have been rarely reported. *A. hybridus* is a widely-distributed, resource-abundant low-cost plant (*Li, 2009*), with high content of soluble polysaccharide plant. It is distributed in southern Shaanxi, Sichuan, Guizhou in China, and also found to be located in Europe, North America and South America. Comparison with some polysaccharides extracted from all kinds of plants, such as *Actinidia arguta* Planch, *Actinidia kolomikta* Planch and Bamboo indicated that *A. hybridus* has relatively high polysaccharide content. However, the research of the polysaccharides in *A. hybridus* is limited; as a result, the resources lag behind in the research and development of phytochemistry.

Therefore, *A. hybridus* was used as material, and the hot water method was applied to extract polysaccharides. Then, analyses of the purification, physicochemical properties and biological activity were carried out, and the results of this study promote the effective utilization of *A. hybridus* and provide a theoretical basis for the extraction process and product development of polysaccharides from *A. hybridus*.

# MATERIALS AND METHODS
## Materials
*Amaranthus hybridus* was obtained from Sichuan Agricultural University and identified by Professor Hui Chen. The materials were placed in the shade to dry and stored at room

temperature. In the study, the shoots of *A. hybridus* was used as material. The mature stage of *A. hybridus* was used for experimentation. The plants were cultivated on the farm of Sichuan Agricultural University. They were sown in April and harvested in September. Nitro blue tetrazolium (NBT), KBr (spectroscopically pure) and a dextran standard were purchased from Sigma (St. Louis, MO, USA). 2,2-diphenyl-1-picrylhydrazyl (DPPH) was purchased from Yuanye Co. (Shanghai, China), Ascorbic acid (Vc) was purchased from Sinopharm Chemical Reagent Co. (Shanghai, China), diethylaminoethyl cellulose-32 (DEAE) was purchased from Whatman Co. (Maidstone, Kent, UK), Sephadex G-100 was purchased from Pharmacia Co. (New York, NY, USA). All other chemicals and reagents were analytical grade.

## Sample handling and calculation
### Sample preparation
The aboveground parts of *A. hybridus* were collected and washed, dried and passed through 80 mesh screens. Samples were crushed by a grinder. For defatting and removing pigment, the powder was placed in a Soxhlet extractor and added to refluxing preheated mineral ether (60–90 °C) at a temperature of 70 °C until the mineral ether was colorless. A step-wise extraction, using solvents having increasing polarity, was used to eliminate pigments, phenolics, carotenoids and other interfering compounds from the plant materials used for polysaccharide preparation. Dry the extracted powder and place it in a drying bottle for backup.

### Preparation of crude polysaccharides
A total of 20 g of the prepared powder of *A. hybridus* in the drying bottle was weighed and mixed with an appropriate volume of distilled water according to the desired solid-liquid ratio. The filtrate was dissolved by hot water extraction and concentrated; then, the resulting solution was mixed with four times its volume of anhydrous ethanol for precipitation for 24 h at 4 °C. The extraction solutions were separated by centrifugation ($5,000 \times g$ for 15 min). Then, the crude polysaccharides were air-dried and weighed. The yield (%) of the polysaccharides was calculated as follows:

$$\text{Extraction yield } (\%) = \frac{m_1}{m_2} \times 100$$

where $m_1$ (g) is the dried crude polysaccharide weight; $m_2$ (g) is the powder weight of *A. hybridus*.

## Purification of crude polysaccharides
### Protein removal from the crude polysaccharides and purification
After precipitation of the polysaccharide powder with ethanol, crude polysaccharides were deproteinated using the Sevage method following *Zhang (2007)*. Sevage (chloroform: N-butanol = 4:1) was added to the polysaccharide solution using a separating funnel to remove proteins. The upper liquid was concentrated using a rotary evaporator (RB-52AA vacuum rotary evaporator); then, the concentrated solution was precipitated by the addition of a fourfold volume of anhydrous ethanol and freeze-drying. The resulting

polysaccharide was prepared in a solution at 10 mg/mL and applied to a DEAE-32 column (*Li et al., 2006*). The polysaccharide was eluted stepwise with distilled water and 0.1–1 mol/L NaCl at 0.6 mL/min. The fractions containing sugar were collected at 10 min per tube using an automatic collector (automatic partial collector BS160A), and the total sugar content was determined and shown in an elution curve.

### Assessment of molecular weight

The average molecular weight of the polysaccharide was determined by gel column chromatography equipped with Sephadex G-100. The column was calibrated with T-series dextran (T-7, 10, 40, 50, 500) as the standard, and the molecular weights of the polysaccharides were estimated by reference to the elution standard curve.

### Component analysis

The neutral saccharide content of the crude polysaccharides was determined based on the phenol-sulfuric acid method (*Cuesta et al., 2003*; *Albalasmeh, Berhe & Ghezzehei, 2013*). The protein content was determined by the Bradford method (*Yang & Wu, 1998*) using bovine serum albumin as a reference. The uronic acid content was determined based on the sulfuric acid-carbazole method (*Fang, Zhao & Zhang, 2011*). Polysaccharide component analysis of *A. hybridus* was adopted by derivatization gas chromatography, and gas chromatography-mass spectrometer (GC-MS). A total of 10 mg of the polysaccharide sample was weighed, then two mol/L five mL trifluoroacetic acid was added and sealed. Place the ampoule bottle in shaker to dissolve the polysaccharide. After the polysaccharide was dissolved, the solution was hydrolyzed 8 h at 100 °C to dissolve it fully. Then, the ampoule bottle was opened, steamed water fully and dissolved in three mL ddH$_2$O. Reduction of 4 h by adding 20 mg NaBH$_4$ at room temperature, neutralization reaction of acetic acid, excess sodium borohydride and drying. Then one mL acetic anhydride and one mL pyridine was added. After shaking and dissolving, acetylation was carried out at 100 °C. After the reaction lasted for 6 h, acetic anhydride and pyridine were dried. The remaining powder was dissolved in CHCl$_3$ and analyzed by GC-MS. Weighing standard sample of monosaccharides mannose, rhamnose, galactose, lyxose, xylose, fructose and glucose, derivatization of saccharonitrile acetate and GC-MS analysis were carried out according to the above methods. An RTX5mx capillary column was used with a flow speed of 1.9 mL/min and sample size of one µL. The initial column temperature was 40 °C and was increased to 290 °C at a rate of 6 °C per minute after 2 min. The temperature was maintained for 10 min, and the temperature of the detector was maintained at 290 °C.

### Ultraviolet and infrared analysis

The crude polysaccharide of *A. hybridus* was dissolved and diluted in an aqueous solution and scanned from 200 to 800 nm with a Ultraviolet (UV) spectrophotometer. Then, one to two mg of sample was crushed and pressed into one mm pellets for Fourier-transform infrared spectroscopy (FT-IR) measurement. For investigating the functional group, the crude polysaccharide was analyzed with a FT-IR spectrophotometer (FTIR-8400S; Shimadzu Co., Kyoto, Japan) over a frequency range of 4,000–400 cm$^{-1}$.
### Scanning electron microscope

The scanning electron microscope (SEM) was used with the ion sputtering deposition method. The polysaccharide powder was subjected to electric conduction and placed in a JSM-7500 SEM platform. The accelerated voltage was 15.0 kV and was increased by 1,000 and 3,000 times.

## Antioxidant activity assays

### DPPH radicals scavenging assay

The DPPH radical scavenging capacity of crude polysaccharide from *A. hybridus* was determined by the DPPH method (*Zhang et al., 2015*). Briefly, two mL of DPPH solution was mixed with a gradient sample solution at different concentrations (zero to two mg/mL) in tubes. The mixture was placed in the dark for 30 min at room temperature. The absorbance was measured at 517 nm. Distilled water was used as the control group, and Vc was used as the positive control. The DPPH radical scavenging ability was calculated with the following equation:

$$\text{DPPH scavenging effect } (\%) = \left(1 - \frac{A_1 - A_2}{A_0}\right) \times 100$$

where $A_0$ is the absorbance of DPPH without sample, $A_1$ is the absorbance of sample and DPPH, and $A_2$ is the absorbance of sample without DPPH.

### Hydroxyl radical scavenging assay

The hydroxyl radical scavenging capacity of polysaccharide from *A. hybridus* was determined using the method reported by Liu et al. Phenanthroline solution (0.75 mmol/L), phosphate buffer saline (PBS) (pH = 7.4, 0.15 mol/L), $FeSO_4$ (0.75 mmol/L), $H_2O_2$ (0.01%) and crude polysaccharide solution of sample (zero to two mg/mL) were stored in 4 °C refrigerator. The reaction system consisted of phenanthroline solution 0.5 mL, PBS 1.0 mL and sample solution 0.5 mL; then, $FeSO_4$ 0.5 mL and $H_2O_2$ 0.5 mL were added. The reaction mixture was incubated in a water bath at 37 °C for 30 min. Absorbance was measured at 510 nm after the mixture was cooled to room temperature. Vc was used as the positive control. The hydroxyl radical scavenging ability was calculated with the following equation:

$$\text{Scavenging rate } (\%) = \left(1 - \frac{A_1 - A_2}{A_0}\right) \times 100$$

where $A_0$ is the absorbance of the sample after reaction with hydroxyl radicals, $A_1$ is the absorbance of the sample, and $A_2$ is the absorbance without $H_2O_2$.

### Superoxide anion free radical scavenging assay

The superoxide anion radical scavenging capacity was determined using the method of *Beauchamp & Fridovich (1971)* and *Zhishen, Mengcheng & Jianming (1999)*. The solution was prepared as follows: PBS (pH = 7.4, 0.05 mol/L), riboflavin ($3.3 \times 10^{-6}$ mol/L), methionine (0.01 mol/L), NBT ($4.6 \times 10^{-5}$ mol/L), and crude polysaccharide solution of sample (zero to two mg/mL). The reaction system contained the four solutions mentioned above and the sample solution 1.0 mL, and distilled water was used in the control

**Table 1 The content of neutral sugar, uronic acid and protein in the crude polysaccharide, respectively.**

| Polysaccharide sample | The content of polysaccharide (%) | The content of uronic acid (%) | The content of protein (%) |
| --- | --- | --- | --- |
| AHP-H | 35.4 ± 0.93 | 28.13 ± 0.35 | 3.14 ± 0.25 |

group. Absorbance was measured at 560 nm after incubating for 30 min. The superoxide anion radical scavenging ability was calculated with the following equation:

$$\text{Scavenging rate } (\%) = \left(1 - \frac{A_1 - A_2}{A_0}\right) \times 100$$

where $A_0$ is the absorbance of sample; $A_1$ is the absorbance of the sample after the reaction.

### $Fe^{3+}$ reducing capacity assay

The reducing capacity of crude polysaccharide from *A. hybridus* was measured by the Prussian blue method (*Hu, 2010*; *Sun, 2010*). $V_c$ was used as the positive control. The solution was prepared as follows: Vc gradient solution (zero to two mg/mL), sample gradient solution (zero to two mg/mL), PBS (pH = 6.6, 0.2 mol/L), potassium ferricyanide (1%, w/v), trichloroacetic acid (TCA, 10%, w/v), and $FeCl_3$ (0.1%, w/v). Then, 2.5 mL of the sample solution, 2.5 mL of PBS; and 1.0 mL of potassium ferricyanide were added to the EP tubes. The reaction mixture was incubated in a water bath at 50 °C for 20 min. When the tubes had cooled to room temperature, 2.5 mL of TCA was added and the samples were centrifuged this mixture at 3,000 rpm for 10 min. 2.0 mL of the supernatant was added into 2.0 mL of distilled water or 0.5 mL of $FeCl_3$. Absorbance was measured at 700 nm after incubation for 10 min. Absorbance was defined as the reducing power.

### Total antioxidant capacity assay

The measurement of total antioxidant activity is able to illustrate the antioxidant capacity when the sample acts on the body. The T-AOC Kit was used in this study. One unit of total antioxidant capacity was defined as when the absorbance of the reaction system was increased by 0.01 per milligram (*Sun, 2013*).

## RESULTS

### Hot water extraction of polysaccharide assay analysis

Response surface methodology (RSM) was used to optimize polysaccharides from *A. hybridus*, and the optimal conditions were as follows: exaction temperature, 92 °C; extraction time, 214 min; and ratio of water to raw material, 43 mL/g. The maximum extraction percent of polysaccharide was 8.29% ± 0.15%. RSM is an optimization method. It takes the response of the system (such as extraction rate in extraction chemistry) as a function of one or more factors (such as extractant concentration, acidity, etc.). The graph technology is used to show the relationship of the functions, therefore, the optimal conditions would be determined.

The chemical components of the crude polysaccharide sample are shown in Table 1. The polysaccharide content was 39.4%, uronic acid content was 28.13% ± 0.35% and protein content was 3.14% ± 0.25%.

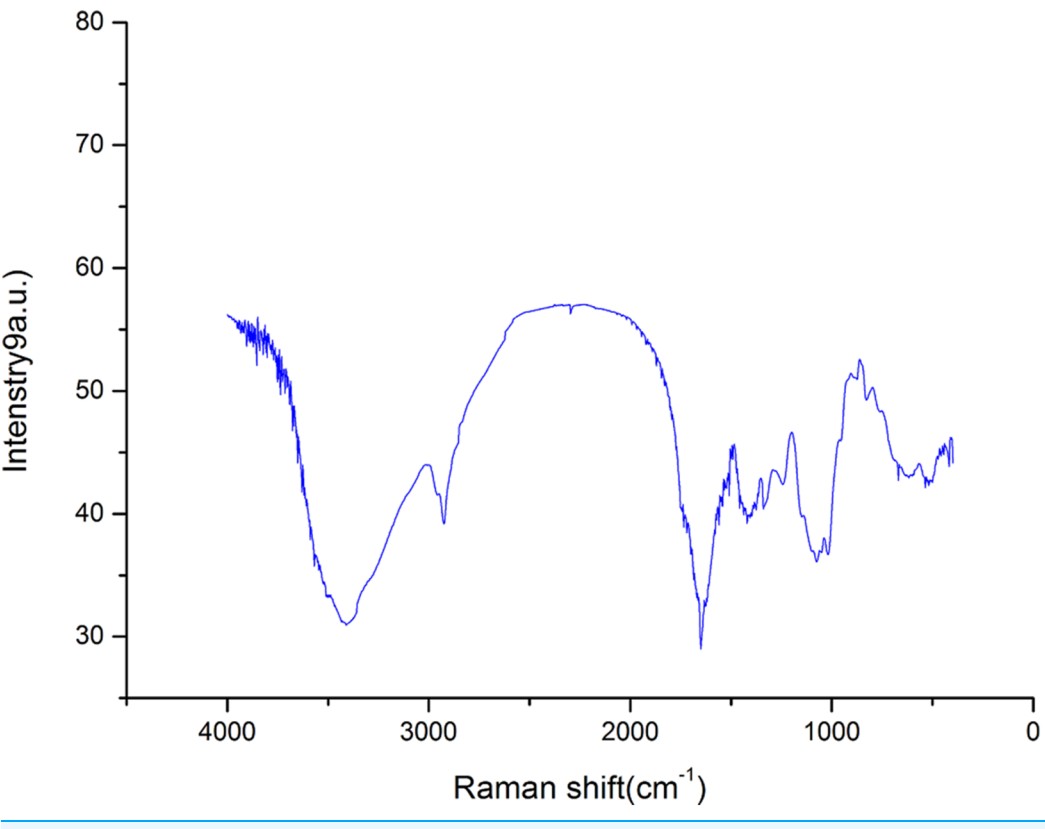

**Figure 1  FT-IR spectra of polysaccharide AHP-H.**

## FT-IR spectroscopy analysis of crude polysaccharide

The FT-IR spectroscopy results of AHP-H is shown in Fig. 1. The spectra showed a broad stretching vibration peak at 3,417 cm$^{-1}$ for the O-H group and a narrow and weak vibration peak at 2,920 cm$^{-1}$. The O-H stretching vibration peak and C-H stretching vibration peak were characteristic absorption peaks of polysaccharide. These peaks revealed that the crude polysaccharide components from *A. hybridus* were carbohydrates.

The asymmetric stretching absorption peak at 1,638 cm$^{-1}$ was attributed to C=O, and the stretching vibration peak at 1,413–1,550 cm$^{-1}$ was attributed to N-H. The two peaks were characteristic absorption peaks of protein, indicating that the sample components contained proteins. The three chemistry components of crude polysaccharide also indicated that the sample contained some protein. The peaks at 1,326 and 1,242 cm$^{-1}$ were possibly due to the stretching vibration of O-H. The peaks at 1,076 and 1,023 cm$^{-1}$ were preliminarily assigned to the stretching vibration of C-O; these peaks were possibly due to C-O-C glycosidic bonds and C-O-H glycosidic rings or may be due to C-O-H and C-O-R of GalA carboxyl groups (*Kacurakova et al., 2000*; *Sun et al., 2005*). The peak at 890 cm$^{-1}$ was the characteristic absorption peak of β-isomers of pyranose, and the peak at 840 cm$^{-1}$ was the characteristic absorption peak of C-H bond of α-isomers of pyranose. AHP-H had an obvious absorption peak at 827 cm$^{-1}$ and a weak absorption peak at 890 cm$^{-1}$, indicating that the main component of AHP-H was α-isomers of pyranose and contained a certain amount of β-isomers of pyranose.

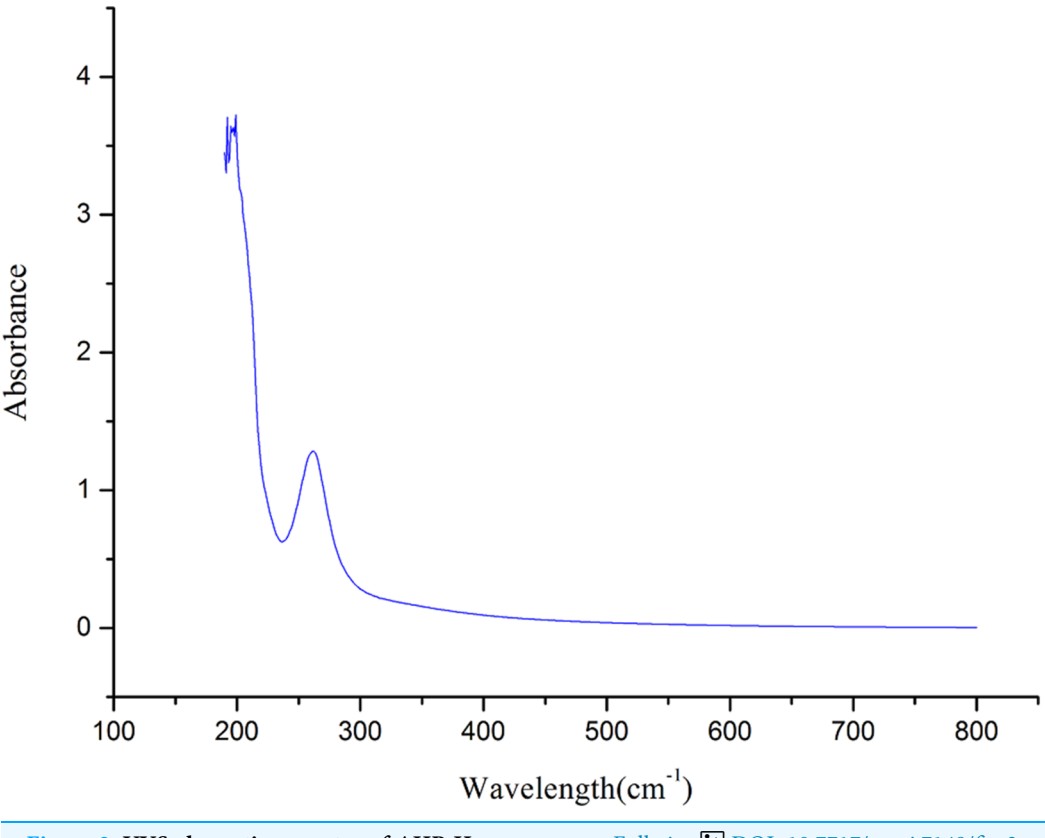

**Figure 2 UVS absorption spectra of AHP-H.**

## UVS analysis of crude polysaccharide

The UVS (*Mao et al., 2006*) of crude polysaccharide from *A. hybridus* is shown in Fig. 2. It is known that crude polysaccharide has absorption peaks at 206 nm. The larger absorption peak was between 250 and 300 nm, indicating that the crude polysaccharide contained proteins and nucleic acid, which is consistent with the FT-IR and of chemistry constitution results. There was no absorption at 620 nm, suggesting the pigment was completely eliminated.

## SEM analysis

The effect on the polysaccharide surface structure is shown in Fig. 3. The hot water method had little effect on the surface structure of polysaccharide since the structure surface was smooth and dense. Compared with hot water method, the surface structure of polysaccharides was affected by microwave-assisted extraction. The surface of polysaccharides extracted by microwave-assisted was uneven and some bubbles appeared. Microwave heating made the polysaccharide heated first from inside to outside, resulting in uneven heat transfer, and the difference of temperature between inside and outside, which resulted in the appearance of bubbles or folds on the surface structure of polysaccharides. It also proved that microwave heating destroyed the surface structure of polysaccharides to a certain extent. These results suggested that the hot water method was gentler than the microwave-assisted extraction, and it was less harmful to the polysaccharide surface

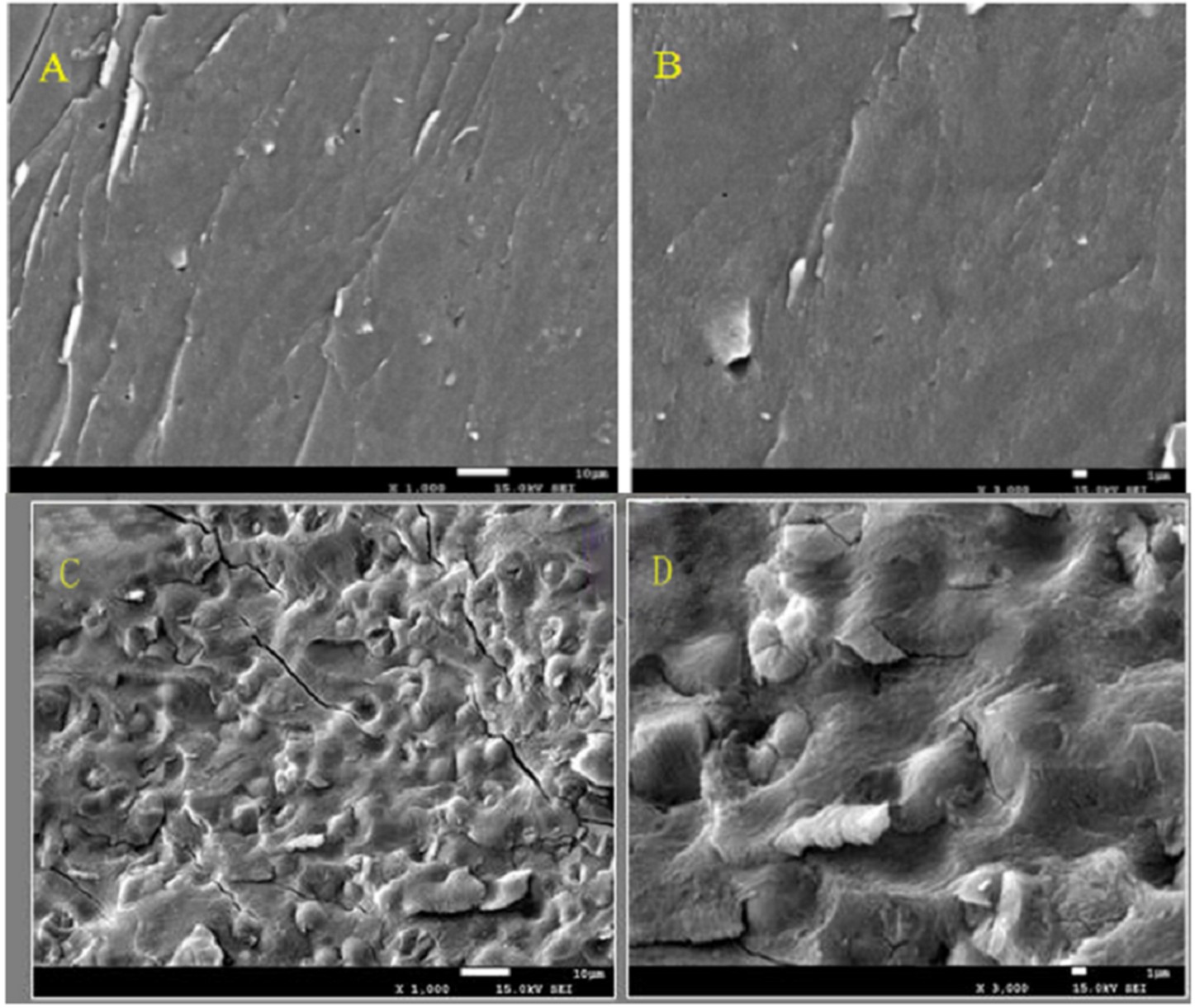

**Figure 3** The figure of Scanning electron micrographs of crude polysaccharide extracted by hot-water method and microwave-assist, (A) AHP-H (1,000×); (B) AHP-H (3,000×); (C) AHP-M (1,000×); (D) AHP-M (3,000×).

structure and was a gentle method of polysaccharide extraction. The structure and function of the substances were adaptable, so it can be inferred that the dense or loose surface structure of the crude polysaccharide of Amaranthus was related to its antioxidant activity. Some experimental results show that glucuronic acid is an effective index of antioxidant activity of polysaccharides, which may be attributed to the presence of electrophilic groups such as aldehydes or ketones in polysaccharides, it can promote the release of hydrogen atoms of O-H bond. Polysaccharide conjugates with high content of glucuronic acid have stronger scavenging effect on reactive oxygen species, indicating that the content of

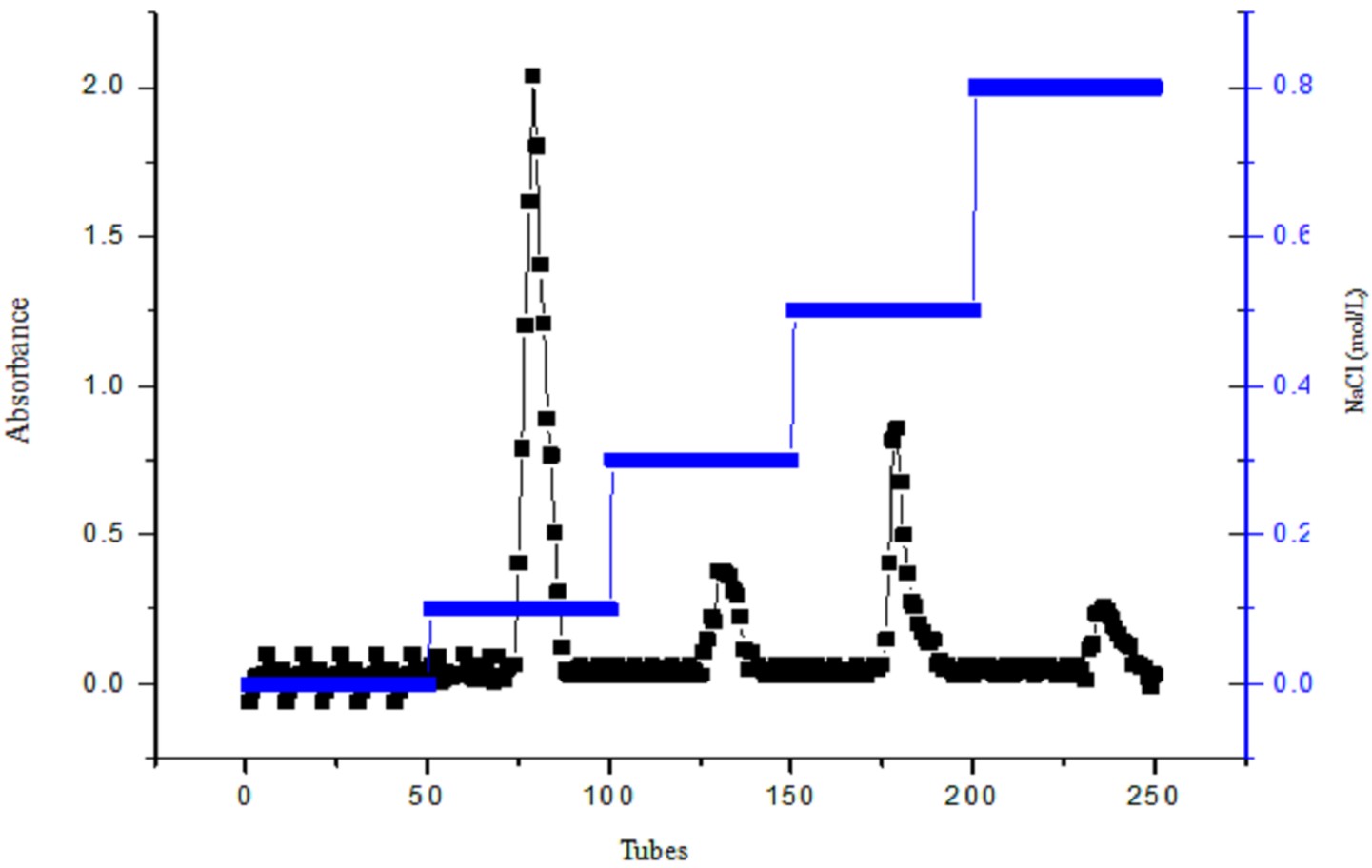

**Figure 4 The figure of DEAE cellulose-32 column chromatogram of polysaccharide, AHP-H.**

glucuronic acid has direct influence on the antioxidant activity of polysaccharides. The dense surface structure was conducive to protecting its uronyl or alcohol hydroxyl groups from being destroyed in extraction, and thus was conducive to its antioxidant activity.

## Preliminary purification of polysaccharides from *A. hybridus*

Diethylaminoethyl cellulose-32 column chromatography was used to purify crude polysaccharides from *A. hybridus* L. The sample was eluted with ddH$_2$O using gradient NaCl solutions of 0.1, 0.3, 0.5 and 0.8 mol/L. The elution curve is shown in Fig. 4. The elution curves of polysaccharides have four peak values, but the ddH$_2$O from tubes one to 50 do not have peaks, indicating that the polysaccharide solution contains acidic polysaccharide instead of neutral polysaccharide.

The elution peak of the polysaccharide presented at 0.1 mol/L NaCl, and when eluted with 0.5 mol/L NaCl, a higher elution peak appeared; yet, the other gradients did not have peaks or their peak values were smaller. Therefore the elution components from 0.1 and 0.5 mol/L NaCl were named AHP-H-1 and AHP-H-2, respectively. The collected components were desalted and dried to obtain the two pure polysaccharide samples.

**Table 2 The content of polysaccharide, uronic acid and protein in the purified polysaccharide, respectively (%).**

| Polysaccharide sample | Neutral polysaccharide content (%) | Uronic-acid content (%) | Protein content (%) |
|---|---|---|---|
| AHP-H-1 | 55.6 ± 0.78 | 40.17 ± 0.62 | 0.24 ± 0.04 |
| AHP-H-2 | 45.7 ± 0.56 | 52.78 ± 0.68 | 0.42 ± 0.12 |

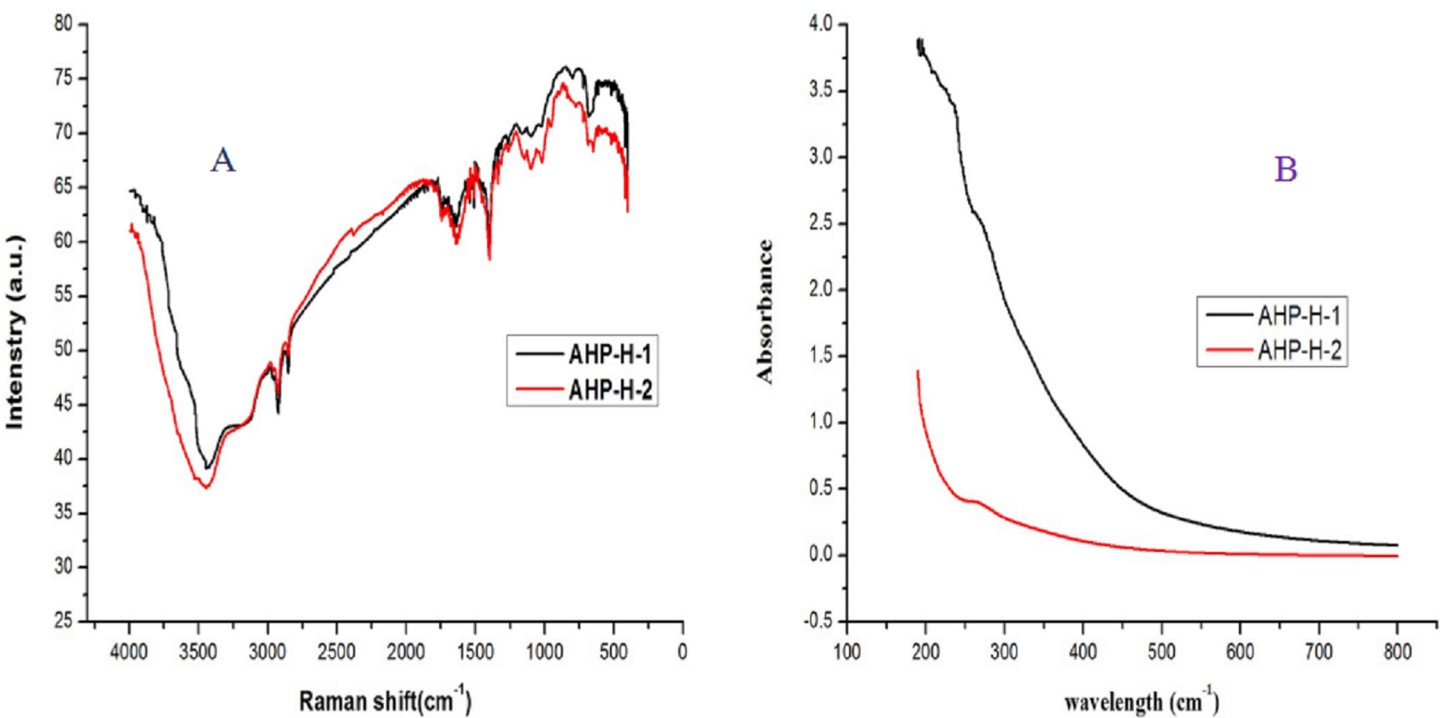

**Figure 5 The FT-IR and UVS analysis of AHP-H-1 and AHP-H-2.** (A) FT-IR analysis of AHP-1 and AHP-2, (B) UVS analysis of AHP-1 and AHP-2.

## Chemical compositions of the purified polysaccharide

The purified polysaccharide contents of the neutral polysaccharides, uronic acid and protein are shown in Table 2. The polysaccharides of *A. hybridus* were acidic polysaccharides. After purification, the contents of the neutral polysaccharides and uronic acid increased and protein content was significantly reduced, which indicated that most of the proteins were removed from the polysaccharide.

## FT-IR and UVS analysis of the purified polysaccharide

The FT-IR analysis results of AHP-H-1 and AHP-H-2 are shown in Fig. 5. The functional groups had no significant differences. The wider O-H stretching vibration peaks were at 3,900 cm$^{-1}$, and the narrower C-H stretching vibration peaks were at 2,930 cm$^{-1}$, indicating that the purified samples were polysaccharides. The C=O asymmetrical stretching vibration peaks were at 1,600 cm$^{-1}$ and N-H stretching vibration peaks were at
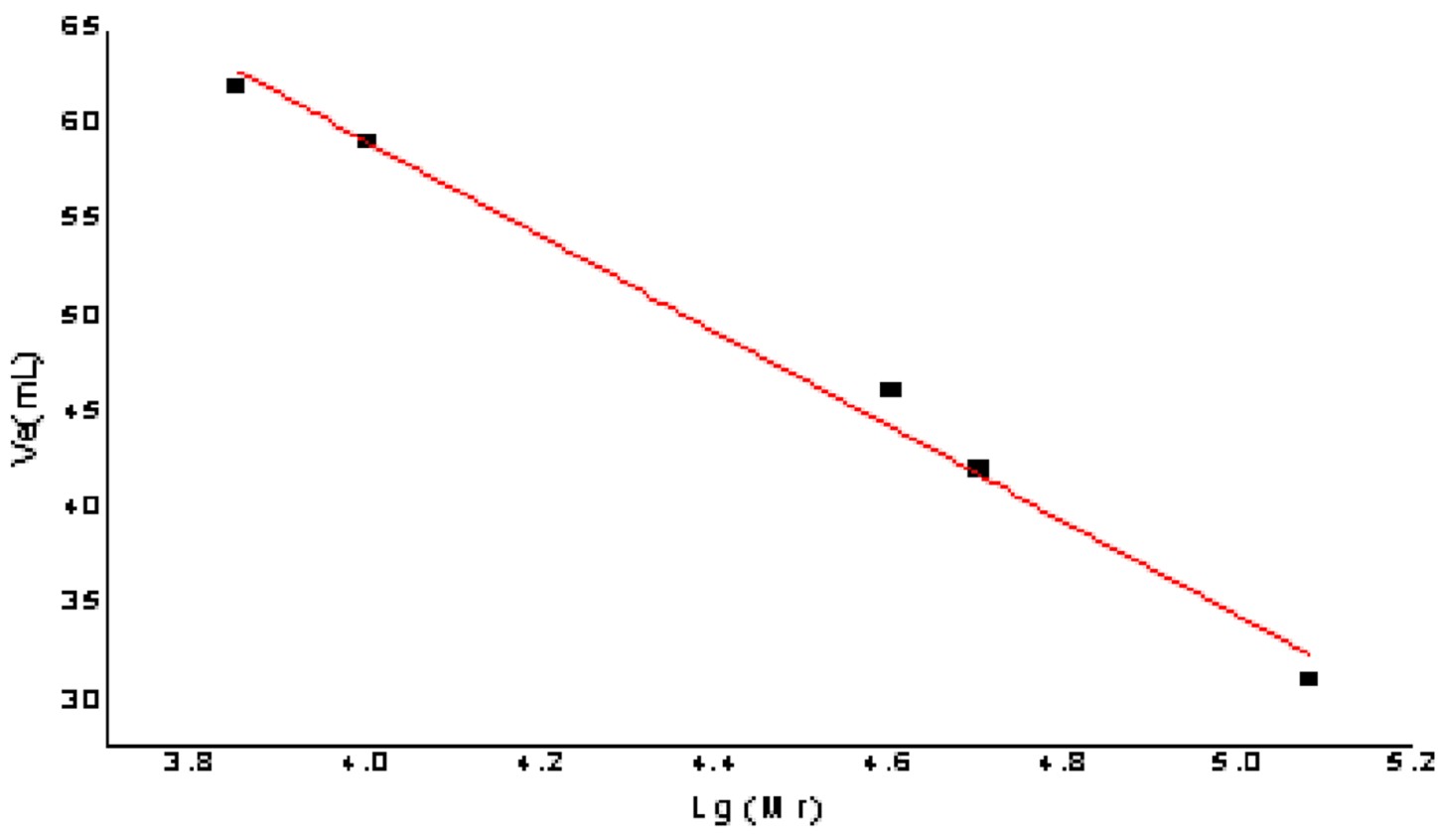

**Figure 6 The figure of standard curve of the relative molecular weight.**

1,500 cm$^{-1}$. FT-IR and UVS analysis demonstrated that nucleic acids and proteins were basically removed, but some protein remained, which indicated there were conjugates of polysaccharides from *A. hybridus* that might be difficult to separate.

## Molecular weight of purified polysaccharides and analysis of the monosaccharide composition

The relative molecular mass of the polysaccharides was measured with Dextran G-100. Drawing a standard curve and taking Ve was taken as the ordinate and lg (Mr) as the abscissa, the calculation formula of the molecular weight standard curve was as follows: Ve = −24.763 lg (Mr) + 158.09, $R^2$ = 0.9913; the formula had a good linear relationship and reliability. The result is shown in Fig. 6.

For determining the elution volume, the two purified polysaccharides AHP-H-1 and AHP-H-2, were eluted according to the standard curve method. The molecular weight is shown in Table 3. The molecular weight of the polysaccharides from *A. hybridus* were 70.795 and 90.3325 kDa, respectively.

Components analysis of purified polysaccharides from *A. hybridus* was carried out through GC-MS, and the results are shown in Fig. 7 and Table 4. There were significant

Table 3 The calculation table of molecular weight.

| Polysaccharide samples | AHP-H-1 | AHP-H-2 |
|---|---|---|
| Ve (mL) | 38 | 35 |
| Lg (Mr) | 4.85 | 4.97 |
| Mr (Da) | 70,795 | 93,325 |

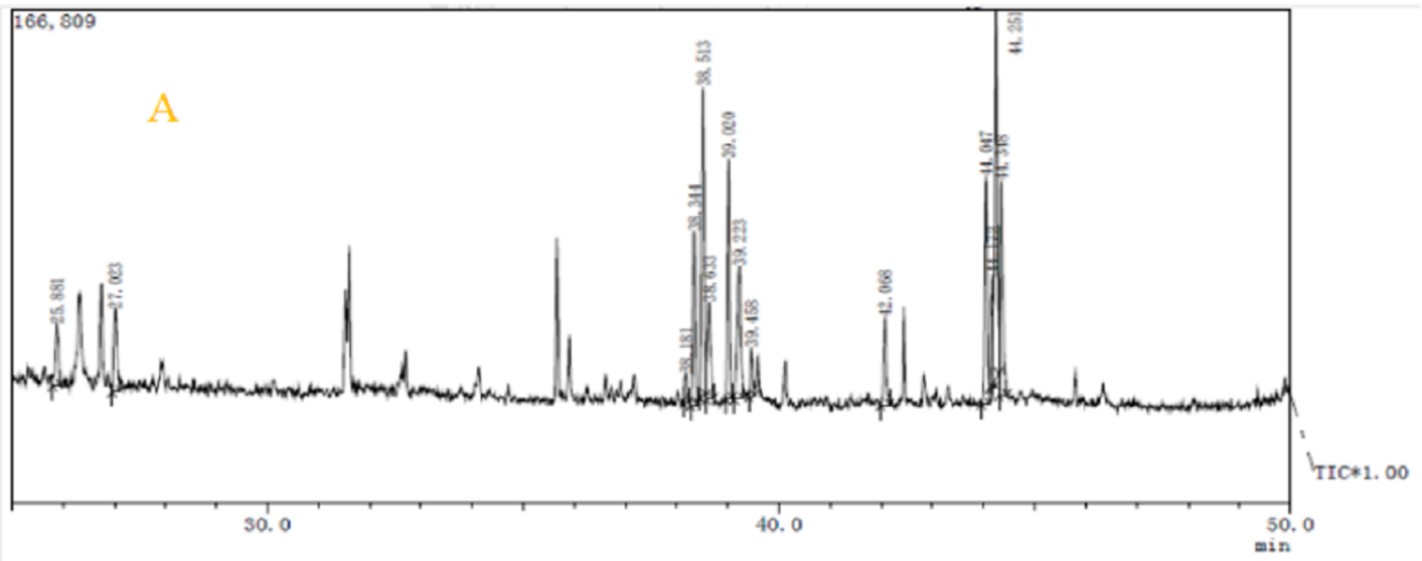

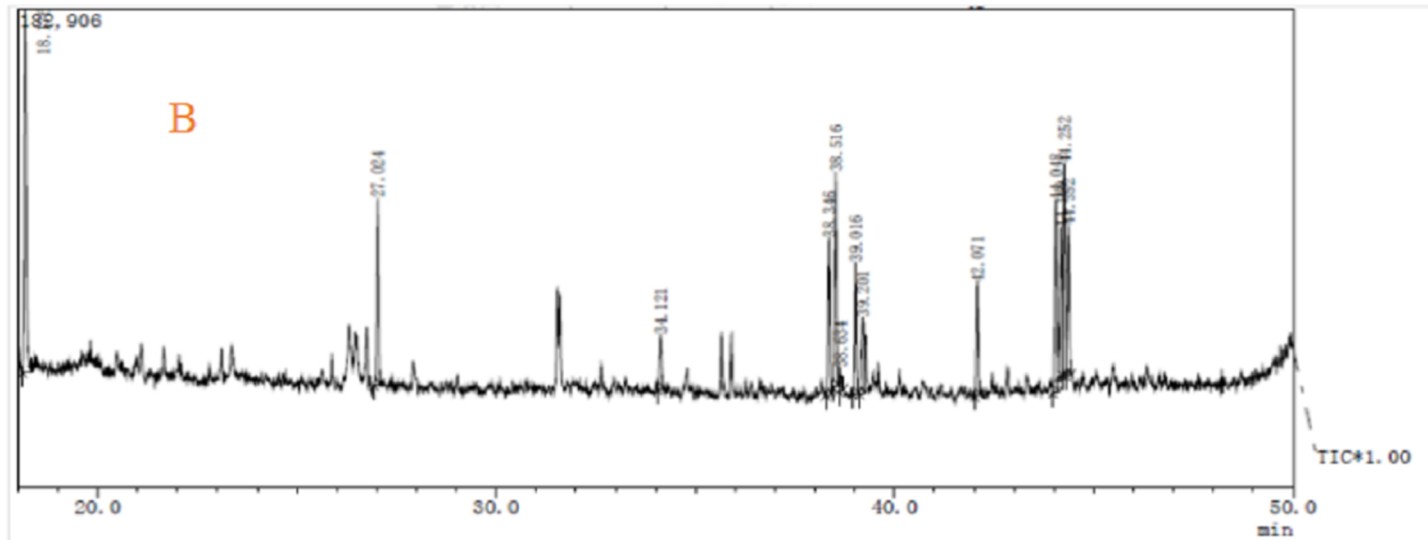

Figure 7 The GC-MS spectrometry of purified polysaccharides of AHP-H-1 (A) and AHP-H-2 (B).

differences between the two polysaccharides. AHP-H-1 consisted of seven monosaccharide components which include mannose, rhamnose, galactose, lyxose, xylose, fructose and glucose. AHP-H-1 did not include xylose and fructose, and the content of glucose was

**Table 4 The monosaccharide composition (%) of purified polysaccharide.**

| Name | AHP-H-1 | AHP-H-2 |
|---|---|---|
| Mannose | 22.2 | 20.41 |
| Rhamnose | 4.29 | 7.02 |
| Galactose | 28.5 | 19.72 |
| Lyxose | 12.99 | 19.34 |
| Xylose | 5.65 | nd |
| Fructose | 5.07 | nd |
| Glucose | 21.36 | 33.43 |

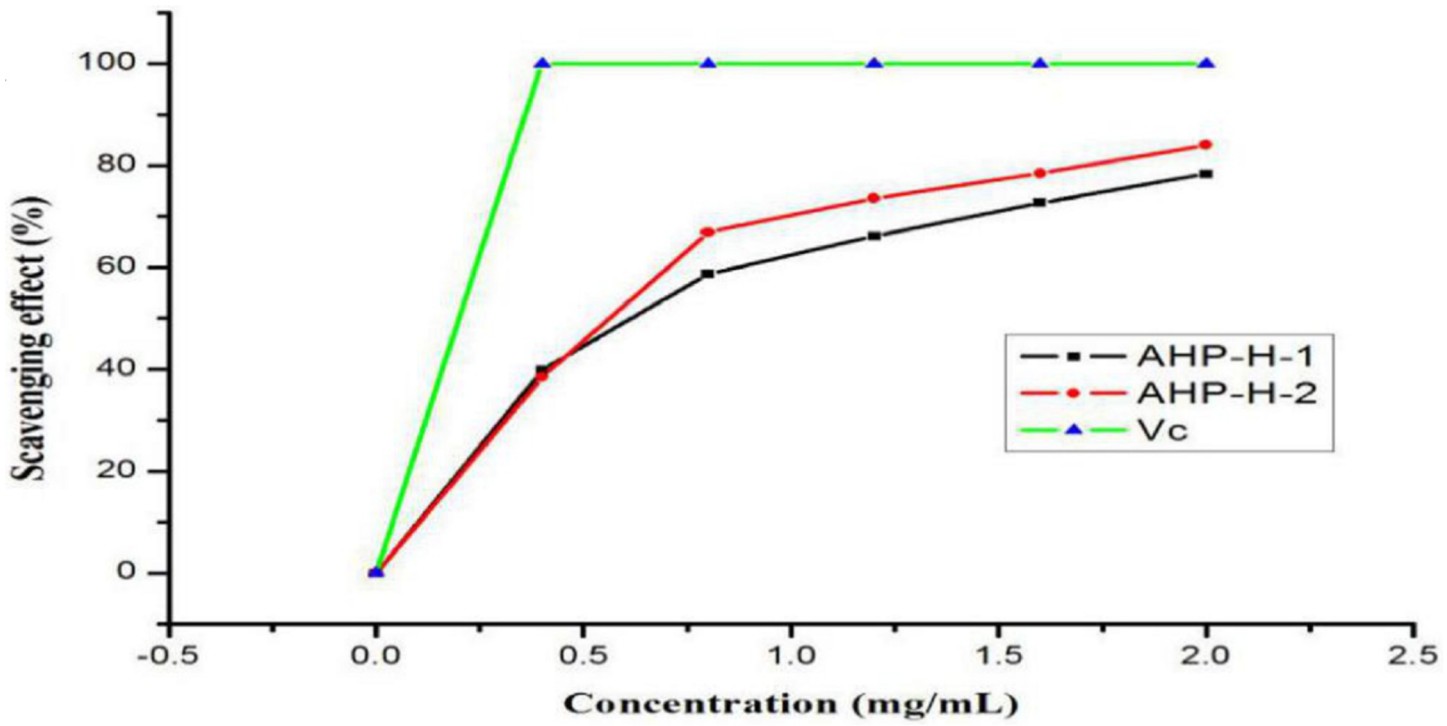

**Figure 8 DPPH radical scavenging ability of purified polysaccharide.**

much higher than that of the other monosaccharide components. Its glucose content was 33.43%.

## Antioxidant activity of the purified polysaccharides

### DPPH radical scavenging activity

The results of the DPPH radical scavenging activity analysis are shown in Fig. 8. All samples had a high DPPH free radical scavenging activity. The free radical scavenging rate of the two polysaccharides basically showed a linear relationship in the range of zero to one mg/mL. When the concentration reached two mg/mL, AHP-H-2 had the highest DPPH radical scavenging activity, the scavenging rate was above 80% and the scavenging rate of AHP-H-1

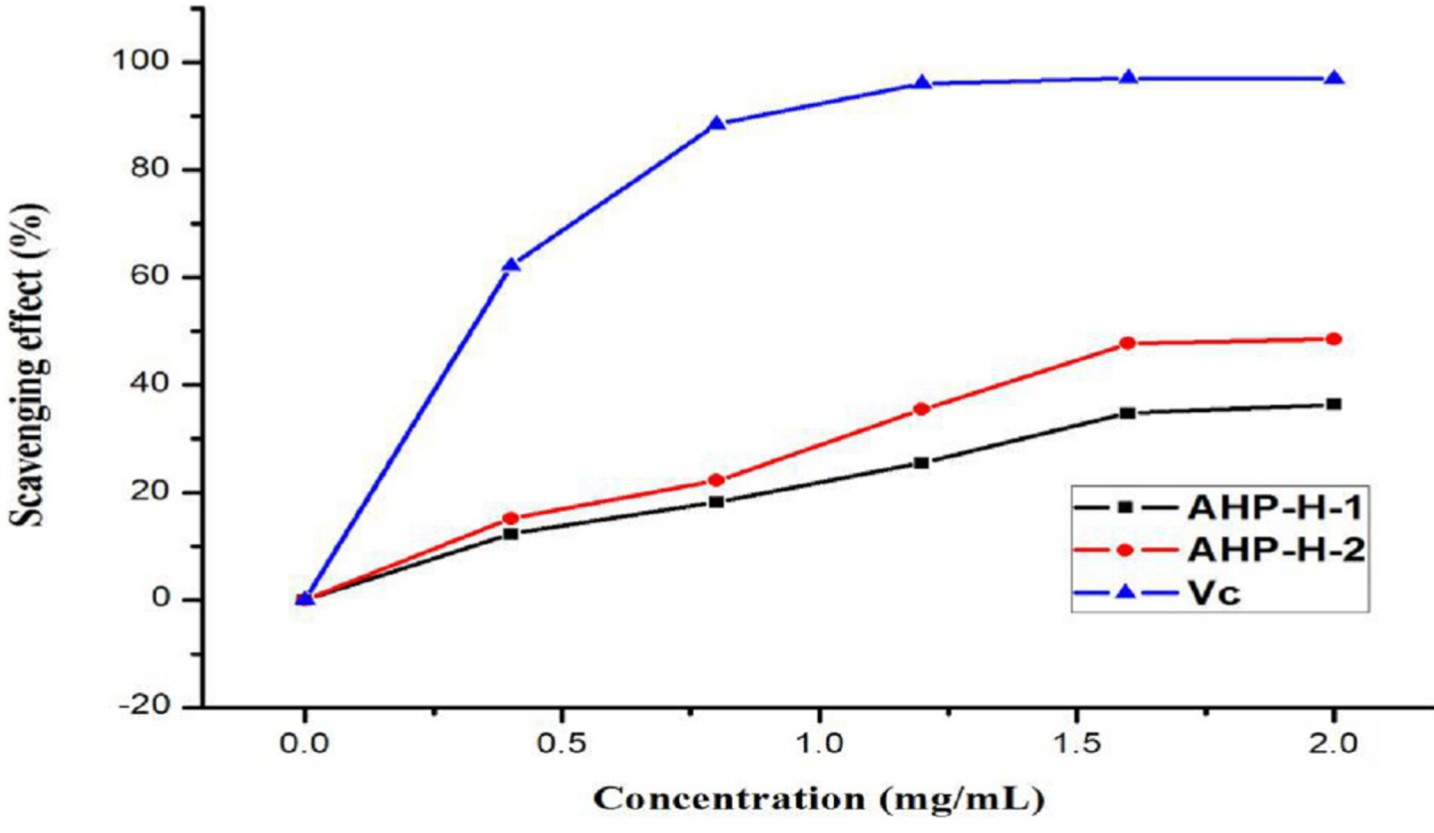

**Figure 9 Hydroxyl radical scavenging ability of purified polysaccharide.**

was close to 80%. The $IC_{50}$ of each component was as follows: AHP-H-1 was 0.908 mg/mL and AHP-H-2 was 0.595 mg/mL.

### Scavenging activity of the hydroxyl radical

The hydroxyl radical is a type of powerful free radical. The scavenging effect of purified polysaccharide from *A. hybridus* was very weak. When the concentration of the polysaccharide solution was at two mg/mL, its scavenging rate was still within 40%. AHP-H-2 had the highest hydroxyl radical scavenging rate at 48.5%. Therefore, polysaccharides from *A. hybridus* were not suitable as powerful antioxidants. The minimum $IC_{50}$ of AHP-H-2 was 2.123 mg/mL. The results are shown in Fig. 9.

### Scavenging activity of the superoxide anion free radical

The purified polysaccharides from *A. hybridus* had a higher scavenging ability than the superoxide anion free radical and had a higher superoxide anion free radical scavenging activity. When the concentration of polysaccharide solution was two mg/mL, the scavenging rate of the superoxide anion radical in solution was more than 60%. AHP-H-2 had the highest superoxide anion radical scavenging rate, close to 80%, and its minimum $IC_{50}$ was 0.999 mg/mL. Superoxide anion radicals mainly destroy organisms' macromolecules, so polysaccharides from *A. hybridus* can be exploited as mild and durable antioxidants. The results are shown in Fig. 10.

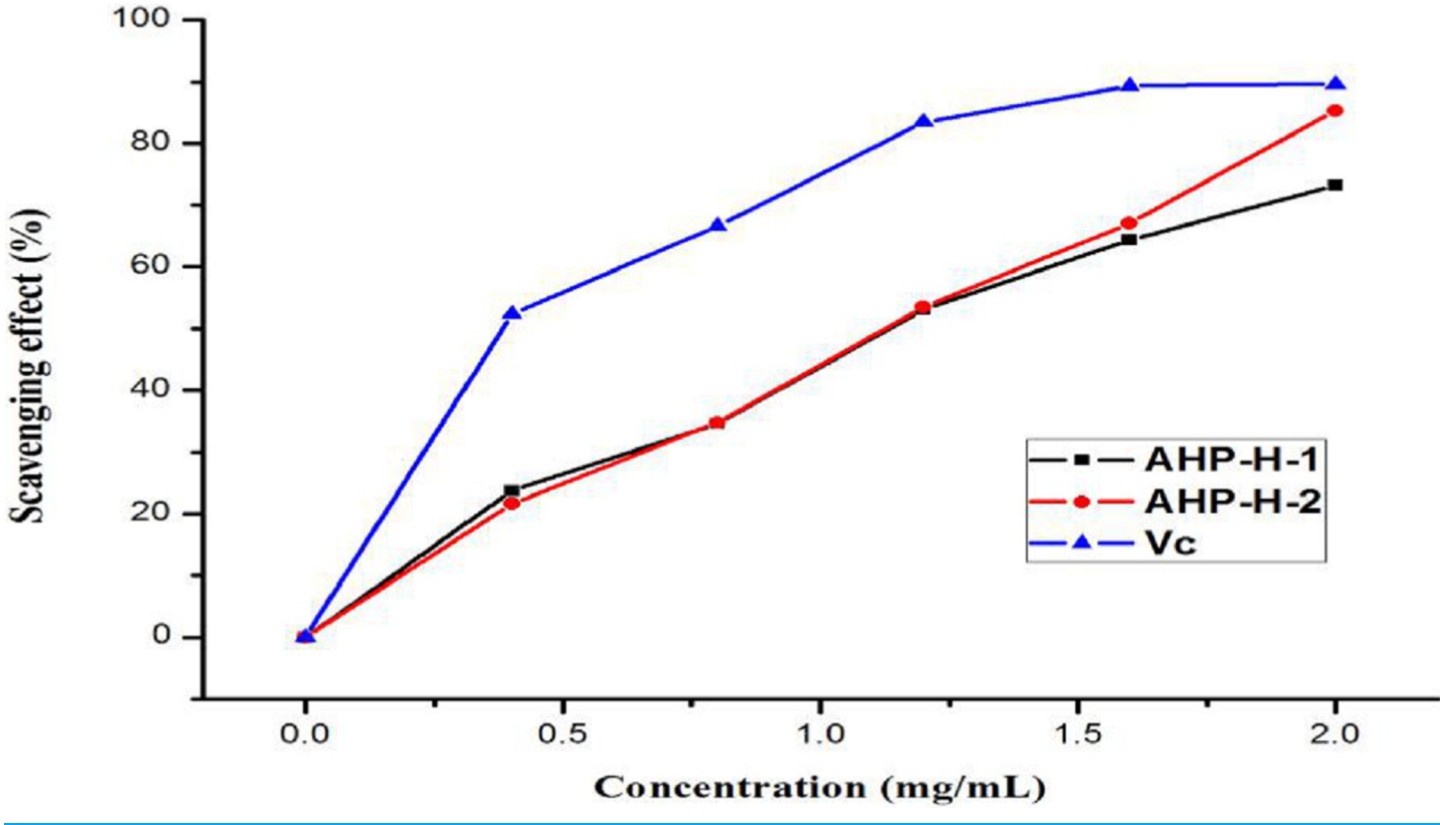

**Figure 10 The superoxide anion free radical scavenging ability of purified polysaccharide.**

### $Fe^{3+}$ reducing capacity results

The reducing capacity of $Fe^{3+}$ is a measure of its oxido-reduction capacity. The reducing power of the purified polysaccharide is shown in Fig. 11. The two types of polysaccharides had a certain reducing power that was dependent on their concentration. When the concentration was two mg/mL, the reducing power of the solution reached its maximum. The strongest reducing powers of AHP-H-1 and AHP-H-2 were 0.695 and 0.918, respectively.

### Total free radical scavenging activity

The T-AOC total antioxidant activity kit was used for investigating the total free radical scavenging activity of the two purified polysaccharides. The total antioxidant capacity per milligram of AHP-H-1 was 2.2 and the total antioxidant capacity per milligram of AHP-H-2 was 6.5. Vc was used as the control, and its total antioxidant capacity per milligram of Vc was 4.5. The total antioxidant capacity of AHP-H-2 was higher than that of Vc, which indicated that the total antioxidant capacity of the polysaccharide with a higher uronic acid content was stronger.

## DISCUSSION

Plant polysaccharides are divided into intracellular polysaccharides, extracellular polysaccharides and cell wall polysaccharides (*Zhang, 2007*). There are differences in the cytoderm of different plants, so there are differences in the extraction conditions from

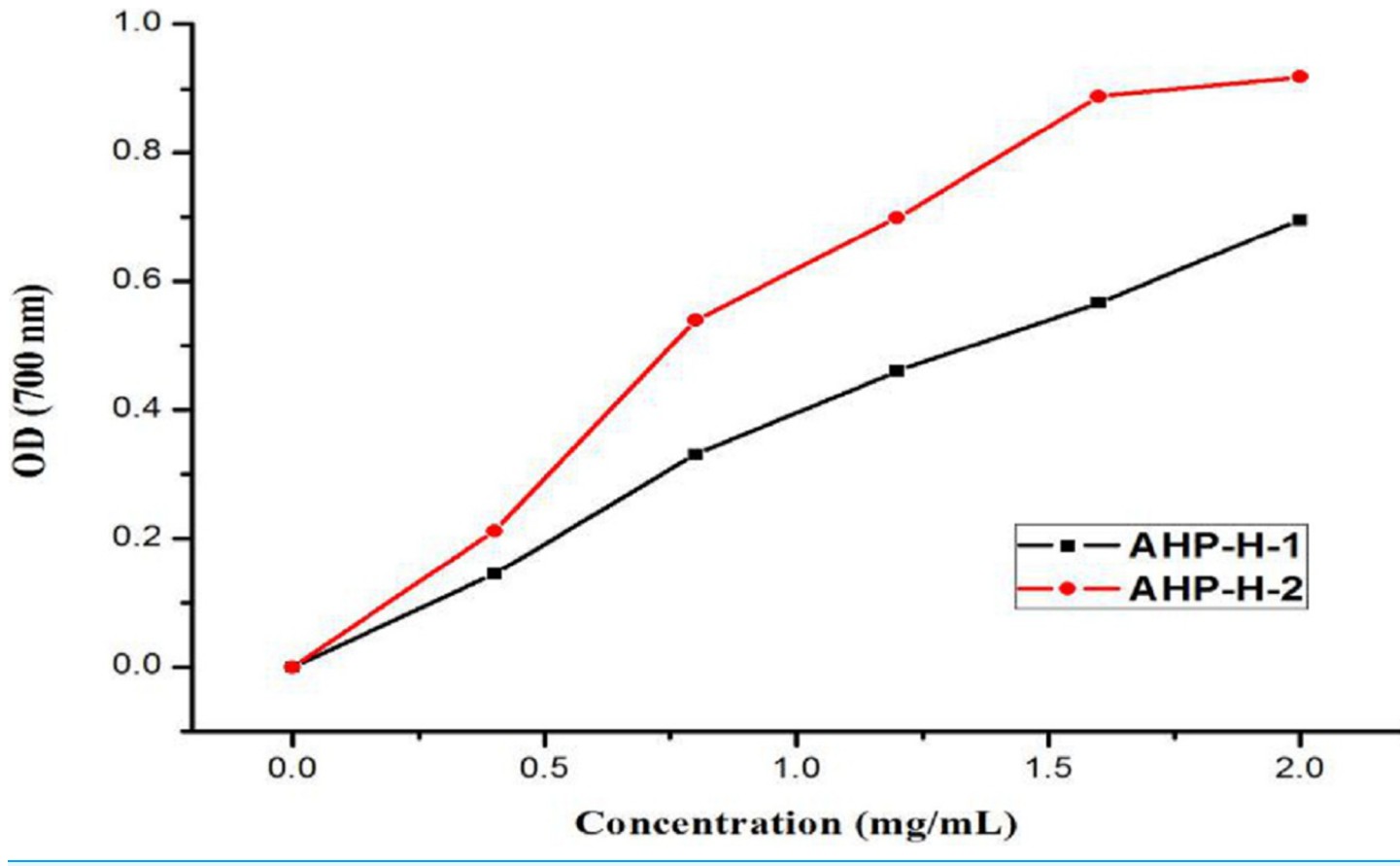

**Figure 11 The reducing power of purified polysaccharide.**

diverse plants, and the extraction conditions of polysaccharides from all kinds of tissues of the same plant are also different, and therefore, the conditions for extracting polysaccharides are different. In the study, polysaccharides from *A. hybridus* were extracted by a hot water method, and when the temperature was lower than 92 °C, the ratio of material to liquid was less than 1:43 mg/mL and the extraction time was less than 214 min, and the extraction rate of polysaccharide increased with the increase in extraction conditions. However, when the extraction conditions were exceeded, the result was the opposite, similar to *Shen et al. (2014)* for *Paris polyphylla* extraction. When the temperature exceeded a certain value, the extraction percent of the polysaccharide decreased with any increasement of temperature, and it is possible that the high temperature causes the dissolution of impurities in the material, which increases the concentration of the solution and reduces the dissolution percent of the polysaccharide, thus affecting the yield of the polysaccharide. SEM observations of the polysaccharides showed that the polysaccharide structure extracted by the hot water method was intact and was not damaged.

It was reported that the amaranth seeds contain 15–19% of proteins with a higher proportion of essential amino acids such as lysine, methionine, and tryptophane, 65–75% of starch, 5–7% of lipids and 3–4% of minerals. They have two to three times higher content of saccharose in comparison to wheat grain. For the neutral sugars, determined

after acid hydrolysis of the sample, glucose is the predominating component, which is in accord with the high starch content of amaranth flours ranging between 62–69%. However, a part of glucose originates from cell wall polysaccharides, that is, cellulose and hexoglycan-type hemicelluloses. Another part together with some mannose was formed during the hydrolysis and reduction steps from the reported saccharose component of amaranth flour. Only about 7% of the neutral sugars accounts for xylose, arabinose and galactose, and presumably they present as hemicelluloses of the arabinoxylan and arabinogalactan types, known to occur in grain of grasses and cereals (*Burisova et al., 2001*). Compared with the Bamboo mentioned in the introduction, they all have glucose, galactose and xylose, and glucose accounts for a large proportion.

The structural characteristics of polysaccharides affect their biological activities, such as chemical composition, molecular weight, type and conformation of glycoside bonds. The in vitro antioxidant activity, reduction ability and free radical scavenging activity of purified samples showed that the antioxidant activity of *A. hybridus* polysaccharides in vitro had no obvious correlation with their monosaccharide composition and proportion, but only with the alcohol hydroxyl group or its chemical molecular structure. Some alcohol hydroxyl groups in polysaccharides can react with metal ions and play an antioxidant role by inhibiting free radical chain reactions (*Qi, 2008*). According to a large number of in vitro antioxidant studies, polysaccharides do have certain antioxidant activities. However, the relationship between antioxidant activity, physical and chemical properties or structural characteristics is not clear, and the underlying mechanism is uncertain. In addition, by comparing a large number of literatures, the results of the conflict were found, because different sources and extraction methods can affect the properties of polysaccharides. On the other hand, there is little information about the antioxidant activity of high purity polysaccharides. Other antioxidants, such as proteins and polyphenols, always exist in polysaccharides in the form of complexes or mixtures. The research on these factors affecting the antioxidant activity of polysaccharides is not very in-depth, nor systematic, which needs further in-depth study. With the deepening of research, the study of antioxidant mechanism of polysaccharides in vitro will be paid more attention.

The results of the experiments showed that the two polysaccharides had strong antioxidant activity, but for hydroxyl free radicals, polysaccharides from *A. hybridus* had not shown good elimination capacity, which illustrates that the polysaccharides are a type of mild radical scavenging substance. The study paved the way for the development and application of polysaccharide antioxidant products.

## CONCLUSIONS

To conclude, the results of the study confirmed that the two polysaccharides purified from *A. hybridus* have strong antioxidant activity, including DPPH radical scavenging activity and superoxide anion free radical scavenging activity. Compared with other antioxidant abilities, the polysaccharides of *A. hybridus* have weaker scavenging ability to hydroxyl radicals. The study was conducted to promote the effective utilization of *A. hybridus* and provide a theoretical basis for its development.

### Funding

This work was supported by the National Natural Science Foundation of China (Grant No. 30671530). The funders had no role in study design, data collection and analysis, decision to publish, or preparation of the manuscript.

### Grant Disclosure

The following grant information was disclosed by the authors:
National Natural Science Foundation of China: 30671530.

### Competing Interests

The authors declare that they have no competing interests.

### Author Contributions

- Yujia Tang performed the experiments, prepared figures and/or tables, authored or reviewed drafts of the paper.
- Yirong Xiao analyzed the data.
- Zizhong Tang conceived and designed the experiments, prepared figures and/or tables, authored or reviewed drafts of the paper, approved the final draft.
- Weiqiong Jin performed the experiments.
- Yinsheng Wang performed the experiments.
- Hui Chen conceived and designed the experiments.
- Huipeng Yao contributed reagents/materials/analysis tools.
- Zhi Shan analyzed the data.
- Tongliang Bu contributed reagents/materials/analysis tools.
- Xiaoli Wang contributed reagents/materials/analysis tools.

### Data Availability

Raw data is available as a Supplemental File.

### Supplemental Information

Supplemental information for this article can be found online at http://dx.doi.org/10.7717/peerj.7149#supplemental-information.

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
