# Peer review of "Extraction of polysaccharides from Amaranthus hybridus L. by hot water and analysis of their antioxidant activity"

_PeerJ, doi:10.7717/peerj.7149_

## Round 0.1 · original submission · Major Revisions

Dear author

Your paper has been assessed by two reviewers and myself as academic Editor.

As you can see below, the manuscript was rated positively, but with major revisions recommended.

Please address all concerns of the reviewers and submit a revised version of the manuscript. Please include a detailed response to each reviewer.

I have been informed that PeerJ is now offering a language correction service for a fee. Please consider it before submitting the new version of the manuscript.

Reviewer 1 ·

Basic reporting

See "General comments for the author"

Experimental design

See "General comments for the author"

Validity of the findings

See "General comments for the author"

Additional comments

The study by Tang et al. describes the extraction, purification and determination of antioxidant activity of two relatively high molecular weight polysaccharide (PS) fractions derived from vegetative tissues of Amaranthus hybridus plants. The study represents a welcome contribution to the existing knowledge of the nutritional components present in amaranth plants, which has predominantly been obtained from the analysis of the highly nutritive seeds produced by the three recognized grain amaranth species (although several studies reporting antioxidant activity in the leafy fraction of various vegetable amaranth species are available; see below). It also presents a comprehensive analysis of the antioxidant capacity (AOC) of these PS fractions, which is unusual and which was based on the use of several analytical assays designed to measure reactive oxygen species of diverse chemical nature. However, there are several aspects of this MS that must be improved in order to be suitable for publication. These are listed below:
Major concerns:
1. The quality of the English language of the MS needs to be significantly increased. Many sections of the MS are difficult to understand, are confusing, or are their meaning is unclear. Examples highlighting these deficiencies are mentioned below.
2. A brief explanation of the chemistry responsible for the antioxidant capacity of carbohydrates should be included in the “Introduction”. The authors might like to include the following information in this respect: Regoli and Winston (1999) Quantification of the oxidant scavenging capacity of antioxidants for peroxynitrite, peroxyl radicals, and hydroxyl radicals. Toxicol. Appl. Pharm. 156: 96-105, and ii) Gutteridge (1984) Reactivity of hydroxyl and hydroxyl-like radical discriminated by release of thiobarbituric acid-reactive material from deoxy sugars, nucleosides and bezoate. Biochem. J. 244: 761-767.
3. In lines 53-54 in “Material and Methods”: Please provide more details about the “specialist” that identified the plants as A. hybridus. A more complete description of the plant material used is also needed: What is meant by “whole shoot”; does it include stem, leaves and panicles? What was the developmental stage of the plants used for experimentation? Where and how were the plants cultivated? In the field, a green-house or a growth chamber? Were they fertilized? How much light were they exposed to? What kind of soil was employed?
4. The molecular weights (MW) of the two PS fractions of A. hybridus seem to be rather large. Please explain how these MWs compare with PSs described in other plants. See Table 3.
5. Lines 101-115: The authors should include a detailed explanation of the methodology employed to identify and quantify each of the sugars-derivatives generated during the GC/MS analysis. These observation applies to Fig. 7, as well. Here, the chromatograms shown are not very helpful. They offer no information to help the reader gain an idea of the PS´s composition. Since no peak is labeled, their identity remains, unfortunately, unknown. This is a pity, considering that evident differences between the two chromatograms are observed.
6. The results shown in Figs. 1 and 2 could be easily eliminated from this MS or shown as supplementary data. This suggestion, on the basis that the information provided is minimal and also redundant. The latter, considering that the authors stipulated that the methodology employed was designed to extract CHOs. Thus, it is not surprising that the IR and UV scan-data yielded spectra that coincided precisely with this premise.
7. Lines 219-222. The sentence is unclear: please re-phrase. I also ask the authors to explain with greater detail how they reached this conclusion. In addition, more information must be included in the figure legend related to Fig. 3 showing the SEM results. To a reader not familiar with this highly specialized topic, the images shown in this figure are abstract and meaningless. In this context, the statement made in lines 320-321 cannot be properly evaluated since no images of a "damaged" PS film were included in Fig. 3 for comparison. In general, Fig. 3 legend is unable to explain the images shown. What are A and B supposed to mean? What are the differences that should be highlighted between the images? What is the reader supposed to learn from these images? The addition of contrasting controls, as mentioned above, would have been helpful.
8. Three peaks are observed in Fig. 4. It is not clear which correspond to the PS fractions?
9. In Fig. 5B, are absorbance values of 4.0 experimentally valid?
10. Data shown in Fig. 6 is not relevant. It should be added as supplementary material. Usually, the use of standards is supposed to produce a straight line if the size-exclusion separation is working properly. Why did the authors omit to show the position where the two PSs eluted in the context of this graph? Mention of the standards used to build it is required.
11. AOC data could be included in a single figure.
12. Title of Table 1 has typographical errors that should be corrected. In addition, the title of Table 2 is misleading, since it seems to refer to the purified PS fractions instead of the crude PS preparation, as stated.
13. Data in Table 4 does not contribute much to determine the origin of the different monosaccharides (e.g., cell wall, hemicellulose, starch, etc.) nor is it connected with data that could permit to gain an insight about the possible structure of these uncommonly large and most probably complex PSs. In this respect, the following study in which the composition and partial structure of the non-starch polysaccharides of amaranth seeds was reported, might be helpful to the authors: Burisová et al. (2001) Isolation and characterization of the non-starch polysaccharides of amaranth seeds. Chem. Pap. 55: 254-260. Do the studies reporting vegetative PS isolated from other plant species, included as supporting information in the “Introduction”, describe partial or total structure of these apparently complex molecules? It is strongly recommended that a comparison in terms of CHO composition, AOC and, perhaps structure, between the A. hybridus and vegetative PSs from other plant species (e.g., those mentioned in the “introduction”) is included in the “Discussion”.
Minor concerns:
1. In the “Abstract”, please be aware that A hybridus is not considered a typical source of grain (mostly due to the pigmented nature of its seeds), although it is believed to be the ancestor of the three recognized grain amaranth species: A. hypochondiacus, A. cruentus and A. caudatus (see Wu and Blair [2017] Diversity in grain amaranths and relatives distinguished by genotyping by sequencing (GBS). Front Plant Sci 8: 1960). Also, please change “…the Amaranthus family Amaranthceae.” to “…the Amaranthaceae family”.
2. In the “Abstract” and subsequent sections of the MS, the change from “aerial parts of the plant” or “above-ground parts” to “shoots” or “vegetative tissues” is suggested.
3. In the “Abstract” and subsequent sections of the MS, the meaning of the abbreviation Vc should be clearly defined when first used. I guess the authors meant it to indicate vitamin C or ascorbic acid.
4. Please rephrase the last sentence of the “Abstract”. Its meaning is unclear and it does not specify how the objectives stated will be achieved.
5. “Amaranthus hybridus” should be added as a keyword.
6. In lines 32-36 of the “Introduction” only one rather obscure and dated reference is included to support the nutritional and nutraceutical value of amaranth. Several recent reviews on this subject have been recently reported and should be included. Among these are the following: i) Montoya-Rodríguez et al. (2015) Identification of bioactive peptide sequences from amaranth (Amaranthus hypochondriacus) seed proteins and their potential role in the prevention of chronic diseases. Compr Rev Food Sci Food Saf 14: 139-158; ii) Venskutonis and Kraujalis (2013) Nutritional components of amaranth seeds and vegetables: A review on composition, properties, and uses. Compr Rev Food Sci Food Saf 12: 381-412; iiI) Rastogi and Shukla (2013) Amaranth: A new millennium crop of nutraceutical values. Crit Rev Food Sci Nutr 53: 109-125, and iv) Caselato-Sousa and Amaya-Farfán (2012) State of knowledge on amaranth grain: a comprehensive review. J Food Sci 77: R93-R104. Also needed are reference to various previous reports describing presence of antioxidant activity in vegetative tissues of various amaranth species, including A. hybridus. These are described in detail in the review by Venskutonis and Kraujalis, 2013, mentioned above. This information could be compared with the results reported by the authors of the MS under revision, in order to determine if the A. hybridus vegetative tissue-related PSs extracted represent a significant improvement, as stated, in terms of AOC
7. In lines 43-44: what is meant by “wide distribution”? Also, the statement “…high [source] content of soluble polysaccharide plant…” should be validated by comparison with other plant PS sources.
8. Line 45: Meaning of the sentence is unclear. Please re-phrase.
9. Lines 56-60: Please include the complete information about the companies that provided the chemicals employed. For instance, change “St. Louis, USA” to “St. Louis, MO, USA”. I guess DPPH was also produced by Sigma, but provided by a local company. Add city and state location for Whatman and Pharmacia.
10. Lines 63-64: perhaps the authors meant “The aboveground parts of A. hybridus were collected and washed, dried, ground and passed through 80 mesh screens”. If the samples were indeed ground, please provide the method used. Was it by mortar and pestle, using a mechanical mill or another method?
11. Lines 66 to 69: Please re-phrase this paragraph: As it stands it is reiterative and difficult to follow. Perhaps start with: “A step-wise extraction, using solvents having increasing polarity, was used to eliminate pigments, phenolics, carotenoids, and other interfering compounds from the plant materials used for PS preparation”.
12. Lines 77-78: Perhaps, the authors meant "yield" instead of "purity".
13. Line 82: It is not clear whether this step was performed before or after precipitation of the PS powder with aq. ethanol.
14. Line 84: Present in the upper phase?
15. Line 90: The authors should be aware that the phenol sulfuric acid method used for carbohydrate quantification is rather unspecific.
16. Lines 106-107: What is meant by “…decompressed moisture completely…”
17. Lines 110-111: please specify how were acetic anhydride and pyridine eliminated. With a stream of N2 gas? ¿Other method(s)?
18. Line 180: What is meant by “Response surface methodology”. A suitable reference related to this methodology should be included.
19. Line 182-183: the authors should be aware that “rate” is usually measured in the context of time. Wouldn´t the use of "percentage" be a more accurate term?
20. Line 186: What could be the origin of the protein “contamination” detected in the PS fractions? Could it be chemically associated with the PSs? Could it include antioxidant enzymes?
21. Lines 240-241: Then, how can the 3% protein content detected in the PS preparations be explained?
22. Line 306: Please define the term "cytoderm". Is it a cell wall, typically of a desmid or diatom? A membrane?
23. Lines 306-309: The authors should notice the highly redundant use of the word "difference" and variations in this paragraph. Please re-phrase.
24. Lines 315-319: The meaning of this paragraph is unclear. Please re-phrase.
25. Lines 323-325: The sentence appears to be contradict previous information. It is also difficult to understand. Please rephrase.
26. Lines 328 and 329: The meaning of “active oxygen redox” and “…which it can be inferred that the uronic acid is favoured over the anti-oxidant” is unclear.
27. Lines 336-340: The meaning of this last paragraph in the “Discussion” is obscure. Please re-phrase in order to clarify the message intended for the reader.
28. Lines 343-345. Doesn´t this statement contradict what was stated above, in the “Discussion” section? Please clarify.
29. Lines 345-346: Is the process cost effective? Is it economically viable? Please elaborate.
30. Lines 357-358: Are these minor contributions enough to endow these co-authors with the right to be listed as a co-author of this study?

·

Basic reporting

although this section complies mostly with the suggested still needs some correction and modifications and there are some questions

Title: In most papers, the word is written antioxidant or not anti-oxidant, it is necessary to correct it unless there is a reason why it is written like that, and so throughout the manuscript

INTRODUCTION
Line 32- such as benefiting the eyes… What type of benefit does it refer to or how is this benefit?
Line 33.. Use etc. instead of so on
Line 34-36. In the introduction refers to the different bioactivities of the amaranthus grains but not the aerial part of it, which is the part that you used in your study, there are already studies on at least the antioxidant activity of these parts, it would be interesting will review them and if necessary, reference them at least in this part of their study
López-García, G., Baeza-Jiménez, R., Garcia-Galindo, H. S., Dublan-Garcia, O., & Lopez-Martinez, L. X. (2018). Cooking treatments effect on bioactive compounds and antioxidant activity of quintonil (Amaranthus hybridus) harvested in spring and fall seasons. CyTA-Journal of Food, 16 (1), 707-714.
Jiménez-Aguilar, D. M., & Grusak, M. A. (2015). Evaluation of minerals,
phytochemical compounds and antioxidant activity of Mexican,
Central American and African green leafy vegetables. Plant Foods
for Human Nutrition, 70, 357-364.
Jiménez-Aguilar, D. M., & Grusak, M. A. (2017). Minerals, vitamin C,
phenolics, flavonoids and antioxidant activity of Amaranthus leafy
vegetables. Journal of Food Composition and Analysis, 58, 33-39.

Line 42-43. A. hybridus is a wide distribution, If it is widely distributed in which parts of the world or in which countries can we find it?

Experimental design

although this section complies mostly with the suggested still needs some correction and modifications and there are some questions


MATERIALS
Line 53. Could you please provide the identification voucher number, the place where it was identified and the geographical location where the amaranthus was collected.
Is the maturity of plants important in the polysaccharide content? If so, please establish the maturity of the plant

Line 69. The extracted material was placed in a drying bottle and dried to a powder. How do you dry the material to get the powder?

Line 78. Please use an equation editor
Line 85. Please set the lyophilization conditions and the equipment used
Line 89. Please mention the type and model of the automatic collector
Line 130. use briefly instead of simply
Line 135. Please use an equation editor

Validity of the findings

although this section complies mostly with the suggested still needs some correction and modifications and there are some questions


Line 273, 280 and 287. IC5O should be IC50 (SUBSCRIPT)
DISCUSSION
Line 337-338. A. hybridus did not have a good scavenging capacity.. should be A. hybridus had not shown good elimination capacity ..

CONCLUSION

The main conclusion comes to the abstract, please try to rewrite it.


Figure 8, the x-axis should start 0 and not -0.5

Additional comments

The results could promote the Amaranthus hybridus consumption as a functional food crop and as an ingredient in food diet.

The discussion has been well designed. However, the manuscript still needs some correction and modifications.

---

## Round 0.2 · Minor Revisions

Dear author,

Your paper has been assessed by two reviewers and myself as academic Editor.

As you could see below, the manuscript has improved considerably, but the reviewers have not yet given green light of full acceptance, but still have some comments. In particular, Reviewer 1 has several issues that you should address, I think you can easily and rapidly incorporate them.

Please address minor issues and submit a revised version of the manuscript. Please include a detailed response to each reviewer.

Reviewer 1 ·

Basic reporting

See below.

Experimental design

See below.

Validity of the findings

See below.

Additional comments

I have carefully examined the revised version of the MS submitted by Tang et al. describing the extraction, purification and determination of antioxidant activity of two relatively high molecular weight polysaccharide (PS) fractions derived from vegetative tissues of Amaranthus hybridus plants. Although some changes suggested by this reviewer were made by the authors, many fundamental questions regarding key aspects of the PS composition and their biological activity remained unanswered or were poorly dealt with. I will again state the salient issues, both major and minor, that need to be properly addressed to reach a suitable level for publication.
Major concerns:
1. The English language quality of the MS is still wanting.
2. The authors were unable to provide a solid explanation of chemistry responsible for the antioxidant capacity (AOC) of carbohydrates, at least in the background information provided in the “Introduction”. Contrary to what is provided in the latter section, that is, that proteins and phenolic acids chemically associated with plant PS provide their AOC, the authors change direction in the “Discussion” section and propose that uronic acids might the responsible for the relatively high AOC produced by both purified A. hybridus PSs. Under these contradictory circumstances, the authors should have made an effort to clarify and support their arguments by adding the appropriate controls, in this case testing if at least commercially available glucuronic acid or pectins had a measurable AOC. Experimental evidence should also be provided to eliminate the possibility that the residual protein still detected in the purified PSs was, inadvertently, the source of their AOC.
3. In “Material and Methods”, the authors ignored the request to provide the name of the “specialist” that identified the plants used in this work as A. hybridus. This introduces a certain degree of uncertainty regarding the identity of the Amaranthus plants used. In this respect it should be noted that A. hybridus plants usually have shorter life cycles than those described by the authors (i.e., from April to September), which introduced the possibility that the authors were probably working with a longer life-spanned Amaranthus species; e.g. A. hypochondriacus. Other information solicited regarding the cultivation of the amaranth plants was either incomplete or not provided at all.
4. The large molecular weights (MW) of the two PS fractions of A. hybridus introduced another contentious aspect to the MS, related to the possibility, even stated by the authors in their rebuttal letter, that such large molecules might be biologically inactive by dint of their large size and, consequently, their inability to freely traverse cell membranes of any kind.
5. The methodology employed to identify and quantify each of the sugars-derivatives generated during the GC/MS analysis remained unclear.
6. The authors mention that, compared with hot water extraction method, the surface structure of polysaccharides was affected by microwave irradiation. It remains unclear why was the method employed compared to the microwave option. Is this method comparable to the water extraction method? Has it been used to extract other plant PSs?
The authors did not respond if the measurement of absorbance values of 4.0 was experimentally valid?
The mass of the dextran polymers used as standards should be included in the figure legend.
I still think that showing all AOC data in a single figure would be a better option.
Minor concerns:
The authors keep ignoring the fact that that A. hybridus is not considered a typical source of grain (mostly due to the pigmented nature of its seeds), although it is believed to be the ancestor of the three recognized grain amaranth species: A. hypochondiacus, A. cruentus and A. caudatus.
In the “Abstract” and subsequent sections of the MS, the meaning of the abbreviation Vc should be clearly defined when first used. I guess the authors meant it to indicate vitamin C or ascorbic acid.

·

Basic reporting

I think the author has completed the review

Experimental design

the author responded according to the objection

Validity of the findings

The author did a good revision work to be able to answer this section

Additional comments

This article shows us other important uses of amaranthus species, it is always good to know how the collection is made and the specific geographical origin of the plant, so that those of us who study this important species have better knowledge and probabilities of comparison between species of the world

---

## Round 0.3 · accepted · Accept

Dear author
I can read that you have addressed all the reviewers concerns. The reviewers comments have been responded adequately. During the production step please correct some typos along the text.
I congratulate you for the nice piece of work, which will add value to PeerJ.

#